# Opportunities and Challenges for the Construction of a Smart City Geo-Spatial Framework in a Small Urban Area in Central China

**Huini Wang [1,2,\*], Ming Zhang [2]** 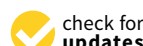 **and Ming Zhong [3]**

[1]  State Key Laboratory of Geodesy and Earth's Dynamics, Institute of Geodesy and Geophysics, Chinese Academy of Sciences, 340 XuDong Rd., Wuhan 430077, Hubei, China

[2]  School of Architecture, The University of Texas at Austin, Austin, TX 78712-1160, USA; zhangm@austin.utexas.edu

[3]  Intelligent Transport Systems Research Center, Engineering Research Center for Transportation Safety, Ministry of Education, National Engineering Research Center for Water Transport Safety, Wuhan University of Technology, 1040 Heping Avenue, Wuhan 430063, Hubei, China; mzhong@whut.edu.cn

\*  Correspondence: wanghuini@asch.whigg.ac.cn

**Abstract:** In 2006, China lunched its first Digital City initiative to build a national geo-spatial framework. Over the past ten years, 511 county-cities benefited from the national initiative with funding and technical resources channeled from the central government. Has the initiative achieved its goals? How has the geo-spatial framework affected local governmental administration, public services, business operation, and the daily life of citizens? What lessons can be learned from the ten-year experience of digital city development? Answering these questions is of important policy, scholarly, and practical interest. The Digital City initiative set the foundation for building smart cities that China's central government agencies and many local municipalities are currently pursuing. A review in retrospect of China's digital city development helps inform future Smart City investment decisions and related policy making in the nation. Lessons learned from the Chinese experience are also valuable to cities in other countries.

**Keywords:** smart city; digital city; geo-spatial framework; e-management and services; Qianjiang City; Hubei China

---

## 1. Introduction

Smart city efforts are bourgeoning around the world. Public agencies in many developed economies and developing countries have launched a variety of smart city initiatives to build up e-governance, aiming to enhance service delivery, increase management efficiency, and improve quality of life of their citizens. There are more than 1000 smart cities in the world that have been launched or under construction [1]. Europe, North America, Japan and South Korea are the leading regions of smart cities. Map vendors such as Google map and Baidu map provide digital spatial information for each city. In addition, large companies such as Cisco and IBM continue to deepen this market, including the 2017 Cisco Announces $1 Billion Program for Smart Cities [2]. These private information services together with government and public applications have covered almost every city in the world. Digital City and Smart City are interrelated but also distinguish with each other conceptually and in practice. Digital City pays attention to the production, accumulation and application of data resources. Smart City focuses on the service design and accommodates users' perspectives. Smart City uses sensor technology and intelligent technologies to realize automatic, real-time operations, and comprehensive perception of urban operations on the basis of Digital City. Cocchia provided a

thorough review of Digital City and Smart City evolution, [3]. Li summarized the achievements in construction and development of the Smart City based on the wave of the Internet of Things (IoT), including smart sensor networks and earth observation networks, and discussed the theories and practice from Digital City to Smart City [4].

China started its first national Digital City initiative in 2006. The main goal was to build the national digital infrastructure, namely the national geo-spatial framework (NGF). Over the past 12 years, 511 county-cities have benefited from this national initiative with funding and technical resources channeled from the central government. In China, the Digital City initiative can be interpreted as Phase I development of Smart City. In the next phase, Smart City's main foci are transitioning from digital infrastructure and virtual representation to information services and intelligence.

Has China's digital city initiative achieved its goals? How has the geo-spatial framework affected local governmental administration, public services, business operation, and the daily lives of citizens? What lessons can be learned from the 12-year experience of digital city development? Are these lessons different from or similar to those observed in other developing countries? Answers to these questions are of great policy, scholarly, and practical interests. Amid a new wave of investments and development in smart cities, a retrospective review of China's digital city development can better inform future investment decisions and related policy making at the national level. Lessons learned from the Chinese experience also have value to cities in other countries.

This paper attempts to answer the above questions by presenting a digital city case of Qianjiang (Digital Qianjiang). Qinjiang is a county-level city of Hubei Province located in central China. The municipal area of Qianjiang consists of the Qianjiang city proper and the towns and rural settlements in the rest of the city. By the end of 2016, the City of Qianjiang had a total population of 962,000, out of which 54.8% was urbanized. This study selected Qianjiang as study case due to the fact that it was the first county-level city in China being selected to pilot test the national Digital City initiative. Digital Qianjiang may be considered as an average case representing many county-cities of similar nature in Central China.

The reset of the paper includes five parts. After the Introduction, Part Two provides a summary review of digital city development in other countries and in China. Part Three describes the study method and introduces the background information about the City of Qianjiang. Next, Part Four presents in detail the design, components, and service application examples of Digital Qianjiang. Part Five discusses the lessons learned from the Digital Qianjiang project and identifies challenges that face digital city development. Finally, Part Six draws conclusions and implications for investment and policy making pertaining to digital/smart city development in China and beyond.

## 2. Digital City Development around the World

### 2.1. In Other Countries

The idea of digital city initially grounded by the early 90s from the America-On-Line cities. Although the United States has never mentioned the digital city, but no matter whether the USGS or American Factfinder are everywhere show the concept of digital city management. In Greece, the Digital City of Trikala (e-Trikala) constitute a framework for digital public services that will benefit both citizens and public administration, shows the procedure by which the Digital City forms an e-Government environment, offering more than administrating services [5]. In Europe, the European Digital Cities Conference has been held annually from 1994 to discuss a wide variety of topics including "the role of cities, towns, and regions in the deployment of advanced telematics solutions within the development of the Information Society". Digital City Amsterdam networking systems, and succeeded to introduce the city metaphor in the regional information services, was built as a platform for various community [6]. In Japan, Ishida etc., used three years to develop a digital city for Kyoto, which builds the interface layer provides 2D and 3D views of the city, the information layer integrates real-time

sensory information related to the city, and the interaction layer assists social interaction among people who are living/visiting in/at the city [7].

Grossner et al. defined a digital earth system as "a comprehensive, distributed geographic information and knowledge organization system", which would be based on participant systems that adhere to an agreed set of protocols and standards for data models, data formats, and metadata to allow it to function as a contributing node in a single, virtual computing system [8]. We can show 80% of the information on human production or life in the digital world. Humans are building a visualization of the world from a global view, so that it will be possible to see in 3, 4, and n dimensions, inside buildings, and what lies underground and underwater [9].

The rapid development of big data, cloud computing, the internet of things and artificial intelligence technology has led a transition from a digital city to a smart city. Smart city, also termed a ubiquitous city, broadband city, knowledge space, and smart community, emphasizes e-services across geographic spaces (cities, states, neighbors, clusters), where information and communication technology (ICT) infrastructures and software applications are fully integrated [10]. In 2010, the United States launched an economic stimulus plan to strengthen smart infrastructure and boost smart applications. The EU has formulated a smart city framework and Singapore proposed to build a "smart country" in 2015 [11]. Smart city developments include a variety of applications by far; these applications may fall into three main types. The first one focuses on environmental protection and carbon emissions reduction. Examples of this type of smart city application include Smart Grid in Vienna [12], Circular Economy in Toronto, and the bike share plan in Paris [13]. The second type aims at improving emergency response, traffic management, and social security—for instance, New York City's system for disaster preparedness [14], London's dynamic congestion control [15], and Tokyo's mobile smartness [16,17]. The third type includes those devoted to industrial applications, for example, in power plants and electric cars (Germany) [18] and clean technologies (Denmark) [19].

## 2.2. Digital City Development in China

In China, the National Development and Reform Commission, Ministry of Housing and Urban-Rural Development, Ministry of Industry and Information Technology and National Administration of Surveying, Mapping and Geoinformation jointly promote the construction of smart cities in light of their respective responsibilities and work characteristics. As a coordinating department, the main task of the National Development and Reform Commission is to coordinate the promotion of smart cities and the overall direction of urbanization reform. As the urban construction department, the Ministry of Urban and Rural Construction focuses on promoting the orderly and sustainable development of urbanization and launched pilot work of smart city in early 2013. The Ministry of Industry and Information Technology, as a department in charge of promoting the construction of urban informatization, has deployed the smart city construction work in three ways: formulation of planning, pilot demonstration, and promotion. The National Administration of Surveying, Mapping and Geoinformation integrates social, economic, humanities and history through the integration of location and time elements and serves the public, enterprises and the government through the construction of a space-time information cloud platform. According to the statistics of China Smart City Development and Research Center, there are 597 smart city pilot projects organized by the National Development and Reform Commission [20], the Ministry of Housing and Urban-Rural Development, the Ministry of Industry and Information Technology, the Ministry of Communications, the Ministry of Science and Technology, the National Standards Commission, the National Tourism Administration, and the State Bureau of Surveying and Mapping. In terms of the construction quantity of smart city, China ranks first with 500 pilot cities, and has formed several smart city clusters such as the Yangtze river delta and pearl river delta.

The first critical step of developing digital city/smart city is to build a national digital infrastructure, known as digital city geo-spatial framework (DCGF) in China [4,21,22]. A DCGF should integrate information from various sources, e.g., the natural environment, socio-economic characteristics,

infrastructure, human settlements, and other related information and enable a wide range of public services [23–25]. From the geomatics perspective, the DCGF should address fundamental and operational issues, including geo-referencing and 3D spatial-temporal modeling, integration of GPS, remote sensing and GIS in mobile platforms, and special data service capabilities in cloud environments [26]. It was expected that the national DCGF to support city planning, management, and operations and to improve public services, governance efficiency, and resource conservation. DCGF implementation took place first targeting prefecture-level cities and then was extended to county-level cities. In twelve years, 334 prefecture-level cities and 511 county-level cities in China received designated funds and constructed DCGF [20].

## 3. Study Method and the Case of Digital Qianjiang

### 3.1. Study Method

This study takes a case study approach to gain insight into the design and implementation of geo-spatial framework for building Smart City in small urban areas in Central China. The selection of Digital Qianjiang for the case study considered two factors.

First, Qianjiang was among the very first group of cities selected nationwide for DCGF implementation. Over the past decade, Digital Qianjiang has undergone a series of processes of digital city construction, demonstration, and promotion, and its DCGF implementation has become a model for digital city development for China's small and medium-sized cities. Nevertheless, Digital Qianjiang did not come without any bumps. There were many institutional, technical, and financial hurdles encountered in constructing Digital Qianjiang. Lessons learned from the Qianjiang case help inform the continuing efforts of digital city/smart city development in other cities of China and abroad.

Second, one of the authors participated in the project for DCGF construction in Qianjiang. Participating in the project allowed direct observations over the design and implementation process as well as the performance of Digital Qianjiang. The technical components of Digital Qianjiang described below were summarized from the original project reports and related documents [27]. Furthermore, when preparing for this manuscript, the authors conducted semi-structured interviews with a number of key individuals who were in charge of Digital Qianjiang. The interview questions were around topics of data acquisition, DCGF platform design, personnel needs, and funding for Digital Qianjiang implementation and maintenance. The information presented here has gone through a public disclosure check.

### 3.2. About Qianjiang

The municipal Qianjiang had an area of 2004 km$^2$ and a total population of 962,000 in 2016. Qianjiang presents a rich set of geodetic, geographic, and socioeconomic conditions that make it a great test bed for initial DCGF construction. Qianjiang sits on the Jianghan oilfield, one of the top 10 oilfields in China, and it consists of 6 state-owned farms and 16 rural communities. Geographically, Qianjiang is surrounded by a number of prominent features; the Three Gorges Dam in the west, the Han River in the north, the Yangtze River in the south, and Wuhan, the Hubei provincial capital, in the east (Figure 1). Qanjiang plays an important role in the regional development initiative, Wuhan Urban Ring, which consists of Wuhan, Qianjiang, and other seven cities around Wuhan. The regional transportation system in Qianjiang includes the China National Highway 318 and the Shanghai-Chengdu expressway, which runs across the city from east to west; two secondary roads Qianjiang and Xiangyue run north to south, and Qianjiang is the pilot city for national road network construction. The Shanghai-Chengdu railway runs through Qianjiang and includes a passenger station and a freight station. Inland waterway transportation moves in all seasons, and the two ports of the Han River in Qianjiang have an annual handing capacity of more than 3 million tons. Last but not least, Qianjiang presents a unique historical factor: Qianjiang is one of the important birthplaces of Chu culture; it was established as a county in 965 CE and upgraded to a city in 1988, and it has thousands of years of stable regional divisions.

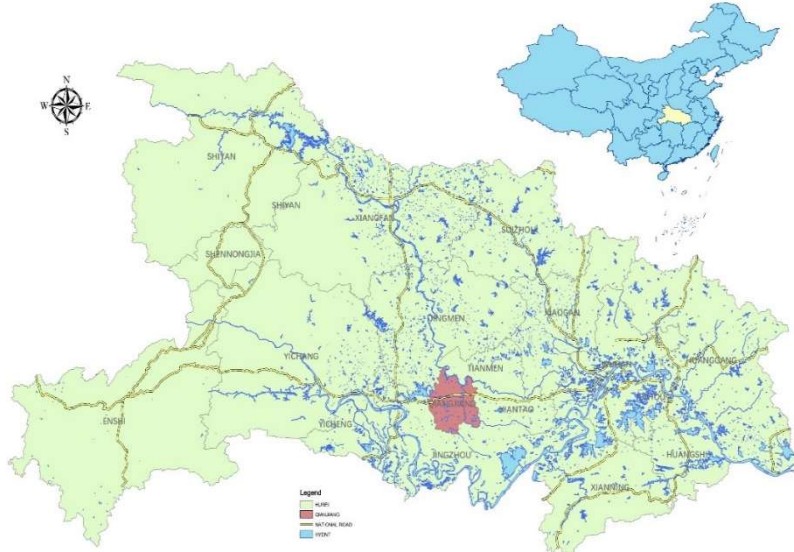

**Figure 1.** Location of Qianjiang in Hubei Province, China.

## 4. Project "Digital Qianjiang"

### 4.1. Project Background

China's National Administration of Surveying, Mapping, and Geoinformation (NASMG) was responsible for DCGF implementation. Figure 2 below shows a layered structure adapted by NASMG in 2005 (Figure 2).

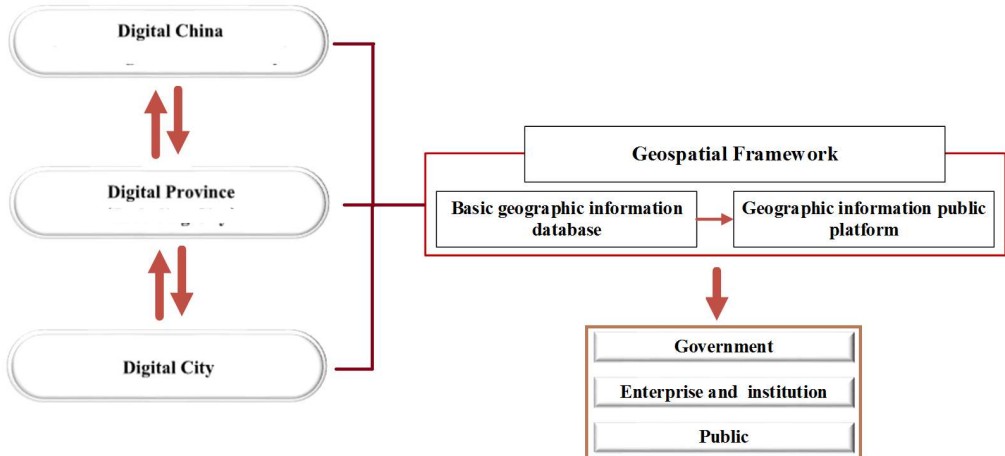

**Figure 2.** Geo-spatial framework of digital China, digital province, and digital city.

### 4.2. Construction Content

Digital Qianjiang geo-spatial framework consists of five parts (Figure 3).

- Basic Geographic Information Data System. This is the core of the geo-spatial framework. It contains surveying and mapping, basic geographic information data, service-oriented product data, management system, and the technical supporting environment.
- Policies, Regulations, and Standards. This system specifies data standards, data exchange protocols, and ensures compliance with relevant policies and codes at the national and local levels.
- Directory and Exchange System. The system serves the function of co-construction and sharing. It includes data inventory and metadata, thematic data, exchange management system, and the technical support environment.

- Public Service System. This system provides an interface for service applications, including maps and data support, online visualization and information services, and the technical support environment.
- Organization and Operation System. This element ensures institutional support for barrier-free operation of the DCGF that involves multiple agencies and stakeholders.

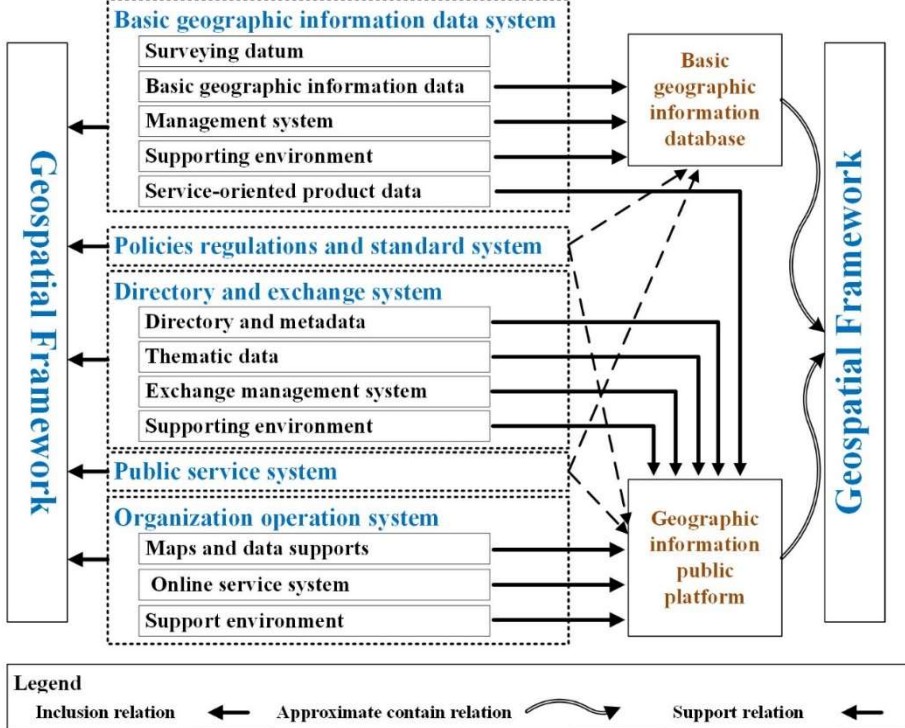

**Figure 3.** Relationships among the geo-spatial framework.

Funding for Digital Qianjiang came from a national-local partnership of three parties, NASMG, Hubei provincial administration of SMG, and Qianjiang City. The total budget for the project was about RMB14 million Yuan with the three partners contributing to the project in the share of 10%, 10%, and 80%, respectively. NASMG paid for system design, high-resolution aerial photographic data, and platform software. The Hubei provincial SMG was responsible for geographic information data services with medium and small scales, database development, and construction of public service platform. The funds from Qianjiang city government paid for GIS datasets, service platform for industry applications, and the hardware (equipment and facilities).

*4.3. Components of Digital Qianjiang*

Digital Qianjiang was completed in four years from October 2005 to July 2009. The project includes four technical components and one non-technological element:

i. Basic geographic information database. The database contains multiple-source and multiple-scale digital datasets covering different parts of the city. They include the 1:6000 aerial photography for the designated urban planning area of 300 km$^2$, the 1:3000 aerial photography for the main urban district of 100 km$^2$, digital line graph data on 1:500 aerial photography for the urban core area of 80 km$^2$, 1:2000 aerial photography from the urban planning area of 320 km$^2$, 1:5000 aerial photography for the urban planning area of 300 km$^2$, and 1:250,000, 1:50,000, and 1:10,000 aerial photography for the entire municipal area of 2004 km$^2$. In addition, the 3D model data on the main urban area of 10 km$^2$ and measurable stereo image sequence data on

the urban area of 576 km$^2$ had been gathered. All these datasets are integrated in the unified surveying and mapping system specified by the national standards.

ii. Public service platform. The platform offers two user versions, the e-government version as intranet for administrative users and the public version for general public users. Each includes specific service and management modules, for instance, online calls, zero code assembly, standard service, customization, and operation maintenance. Furthermore, the platform enables data integration and sharing to meet various needs of government agencies, industrial sectors, and individuals.

iii. Application and demonstration modules. The Digital Qianjiang project offers a number of application and demonstration modules for users in both the public and the private sectors. For example, one module provides support to the planning and management for Qianjiang's economic development and urban and rural development agencies. Another module offers digital information services to Jianghan Oil-Extracting Company. Still, another module serves the needs of local Centers for Disease Control and Prevention for epidemic disease prevention and control.

iv. System support. This component ensures a digital network environment that meets application needs and complies with various technical and security requirements. It includes four sub-systems of hardware and software for network, server, storage, and security purposes. The system support component offers support layers under three operating environments: a secured environment designated for handling classified data and documents, governmental affairs, an intranet environment for government internal affairs, and an internet environment accessible by the general public.

v. Personnel training and talent recruiting. The Digital Qianjiang project sets up a specific program to train and recruit needed talents for the operation and maintenance of the system after project completion.

### 4.4. Application Examples of Digital Qianjiang

During its 12 years in development, Digital Qianjiang demonstrated significant benefits to local government, business, and residents. Below we introduce three examples of application modules [28].

### 4.4.1. Qianjiang Information System for Epidemic Disease Prevention and Control (QISEDPC)

This module aimed to help local health agencies manage and control schistosomiasis disease impacts. Qianjiang, along with a number of its neighboring cities and counties, was among the most affected areas by schistosomiasis disease in China in the 1960s to 1970s. While the infection rate and impacts have reduced significantly lately, local residents and animal stocks remain at high risk to schistosomiasis infection. In Qianjiang, out of total 334 villages, 299 of them are on the watch list of epidemic villages, presenting arduous work loads to local health agencies and governments for schistosomiasis and other epidemic disease control and prevention [29].

QISEDPC offers five functions: data browsing, epidemic investigation, epidemic prevention and control, epidemic statistics, and outbreak alert and response. It can locate medical resources, search epidemic information, browse the disease source distribution and epidemic monitoring statistics, and make decisions on the epidemic information, providing scientific technical means for medical treatment, scientific control, monitoring, and management for the government and the health department. Since it went online in full operation, QISEDPC has helped local health agencies greatly improve their work in the prevention and control of schistosomiasis. Staff in the health department can visualize historic data on the spatial distribution of schistosomiasis incidents and quickly analyze/map the infection rate that in the past came in tables only or in paper maps to the best. Experienced health staff could make intuitive, quick assessments of infection trends without getting into sophisticated statistical modeling and come up with quick-response plans for prevention and treatment. For instance, the Town of Haokou had been rated in the high-risk zone in the past. After displaying and analyzing infection

data along with demographic and geoinformation data in 2006 and 2007 in QISEDPC; the local health workers and government agencies prepared and implemented a targeted plan for prevention. The 2008 assessment showed Haokou having reduced risk level from high to moderate [28]. In another example of Fengjiatao Village (Figure 4b–d), schistosomiasis infections elevated from light in 2006, to severe in 2008. Local health workers used QISEDPC to overlay water surface coverage including rivers, trunk channels, branch channels and large paddy fields, which provide favored conditions for the spread of schistosomiasis, with incident spots. Based on the spatial analysis they came up with a treatment plan. According to the statistics from 2006 to 2014, the residents' schistosomiasis infection rates were 18.31% from blood test and 5.23% from stool examination in 2006; these dropped to 3.78% and 1.14%, respectively, in 2014 [29]. The schistosomiasis infection rate in cattle dropped from 3.27% in 2006 to 0.00% in 2014, and the number of acute infections fell to zero [29]. QISEDPC enabled quick identification of high-risk hotspots and effective treatment plans at a cost (in terms of personnel and medication) much lower than before.

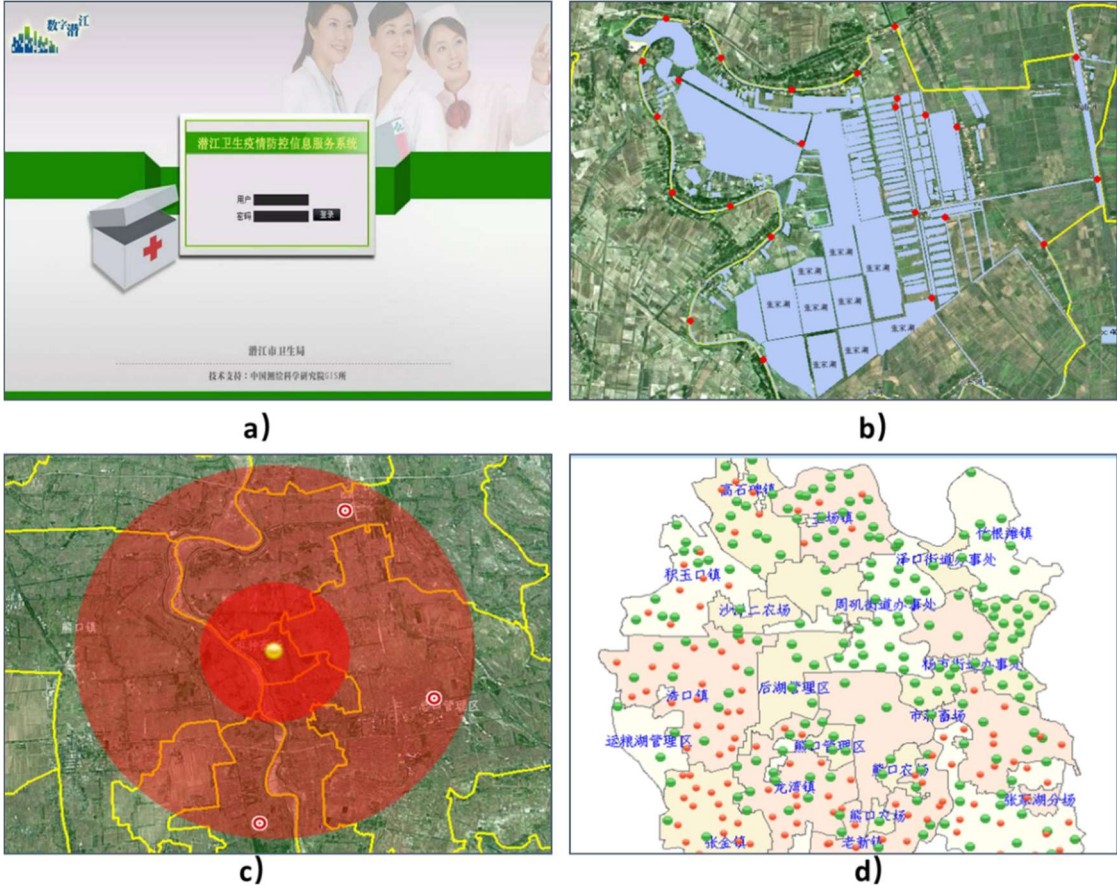

**Figure 4.** Health epidemic prevention and control system (**a**) System login; (**b**) Distribution of affected areas; (**c**) Analysis of epidemic buffer zone; (**d**) Insecticidal drugs allocation.

4.4.2. Digital Information Service System for the Ground Engineering of Oil-Extracting Company (DISS-GEOEC)

Jianghan Oil-Extracting Company (JOEC) is affiliated with Jianghan Oilfield, part of China's state-owned Sinopec. DISS-GEOEC has played a critical role since its inception in improving JOEC's operating efficiency, incident detection, and cost savings. DISS-GEOEC can visualize information on oil well/station locations, equipment and model parameters, dynamic and static data records, and construction and management schedule for new projects (Figure 5). JOEC has utilized DISS-GEOEC to monitor their well operations real-time; this real-time monitoring capability helps shorten troubleshooting times for oil wells and effectively improves the safety feature of the equipment.

For instance, aided by DISS-GEOEC, the Wuqi operating area of JOEC discovered and handled more than 70 abnormal well stoppages and more than 90 process failures in 2009. In addition, DISS-GEOEC supported more than 20 times of pipe network overhaul and 5 times key station renovations; the total work time needed for the renovations was reduced to 10 days, which typically took two months in the past without DISS-GEOEC [30].

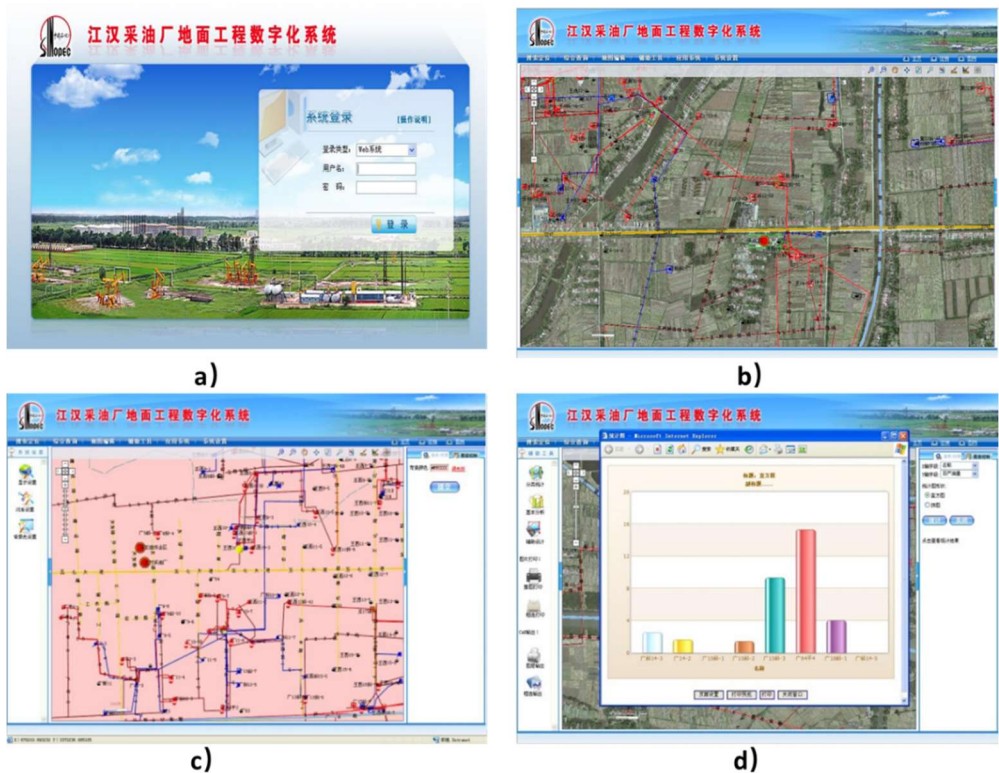

**Figure 5.** Jianghan oilfield oil-extracting factory (**a**) System login; (**b**) Pipeline diagram; (**c**) Well distribution diagram; (**d**) Statistical analysis of well daily output.

### 4.4.3. Improved Public Services and e-Governance

For the public, the role, and the meaning of Digital Qianjiang in urban construction are more intuitive and significant, and ordinary citizens can feel it. For example, the Digital Qianjiang public service system provides various real geographical space applications such as online 2D and 3D mapping of large cities, map browsing, geographic measurement, interest in dots, geographic information query, route query, and information visualization for the public, and provides high-quality real-time service experience of online map and meets the needs of residents for traveling, residence, purchasing, and entertaining.

According to statistics from Qianjiang city's financial department, Qianjiang's economic indicators have undergone a great breakthrough via application of Digital Qianjiang construction achievements. More than RMB 11 million has been saved in 2009.

For the E-government affair intranet, Digital Qianjiang provides geo-spatial information service for demonstration and applications department through an e-government network. While connected on this network, all departments can use this platform for free. Every department can access real-time geo-spatial information on a large scale and overlay its own thematic information. It thus forms a professional service system and realizes the co-construction and sharing connectivity distribution service mode, which can fundamentally solve the problems of the lack of a unified data standard and shared information because each department does things its own way and, and because repetitive construction and can greatly reduce administrative costs, improve work efficiency, promote the

service level of government departments, and strongly promote the rapid and healthy development of Qianjiang's economy.

## 5. Lessons Learned from the Qianjiang Case

Since 2006, the State Bureau of Surveying and Mapping has carried out the construction of a digital urban geo-spatial framework. The primary problem to be solved is the problem of urban geo-spatial data with and without. The most important purpose is to use the city as the unit to collect comprehensive spatial data at the end. And gradually gather up to become the basic geo-spatial data at the provincial and national levels, in order to achieve full coverage of geographic information data from the city's large scale to the national level. The data coverage of the underlying geo-spatial work is similar to the multisource data collection of the USGS. On the basis of realizing the coverage of spatial data, the digital urban geo-spatial framework is also committed to let the various departments of the government use the spatial data reasonably and effectively to carry out urban management work, and realize the spatialization and digitization of urban management work. The Digital Qianjiang Geo-spatial Framework carried out construction work in accordance with this goal, and also carried out socialized services of spatial data in the form of pilots. Although the present state of China's digital city has seen progress in its building of space frame, it remains a great distance from other countries' digital city construction owing to its late beginning, poor initial conditions, and many problems during its development.

### 5.1. Data Acquisition

Data Fragmentation and Inconsistency. The very first challenge facing Digital Qianjiang project was data integration from different sources. The public sector was nearly the only source of geographic, socioeconomic, and demographic data. However, these datasets came in a variety of formats and metrics; most of them are inconsistent in spatial coordinates and software platform. Other datasets existed in text or hardcopy only. Digitizing these datasets and integrate them in one consistent system involved tremendous efforts. most data of industry application departments are textual data, the main problems are the spatialization of such text data and the construction of industry thematic data. For example, in Qianjiang's case, all the data in the health sector are textual and need to be spatially processed according to the geographical names and addresses. Third, every city's basic geographic information database is separate because of their different data formats, software platforms, database platforms, and professional technical frameworks, which have resulted in a complex of disunion, closeness, and non-interaction. Various data also cause obstacles in interconnection and seamless linking.

Perfect and unified standard system and sharing mechanism and standard construction. The National Administration of Surveying, Mapping, and Geoinformation, along with relevant departments, should seek uniform legislation regarding the production and updating of cities' data and set up an effective means of data production, updating, and security. Although a variety of construction standards of the digital city and geo-spatial framework have been launched by the National Administration of Surveying, Mapping, and Geoinformation and relevant departments; the standard system is still imperfect, and criteria vary. It is necessary for the construction of the digital city to establish unified criteria. With the development of information technology, other departments have various requirements of the basic geographic database, and the digital city's related standard system must still be improved and upgraded. A sound information management system, information development planning, and service standards should be established in areas like the digital city's construction framework, system construction, and service pattern. They should also standardize and integrate construction of geo-spatial frameworks and intensify the top-level design of the digital city.

### 5.2. Platform Service and Application Service Extension

A modern platform service has problems when linked with all kinds of systems in a digital city. The geographic information public service platform currently lacks versatility and applicability and is not practical. Many cities have built their own geographic information public service platforms, but they cannot assist other systems in a digital city, like e-government, e-commerce, and intelligent transportation systems. The reason for their inefficiency is their different data management modes, development languages, deployment platforms, and external interfaces. Furthermore, a low level of intelligence and limited auxiliary decision-making ability make it difficult to maintain effective operation, application, and utility.

Fundamental software to enhance openness and interactivity should be strengthened. Research on publicity and the interaction of the spatial information service should be reinforced. The ability to develop web space service components and create universal interfaces to improve the application in space frame should also be strengthened. In addition, the application system and construction must be improved to potentiate links and alternation between the geo-spatial framework and the digital city's system and software.

### 5.3. Talent Issue

Many subjects are involved in the digital city, so it is important to have many versatile and talented professionals who are proficient with computers, GIS, the internet, software, management, society, humanity, and other subjects. In the construction of digital Qianjiang, many geographic information technical workers built the geo-spatial framework within a year, however, it almost took three years to make substantive progress for industry applications. The main reason is the informationalized level of the industry application department lagged and lacked personnel specializing in information technologies. The digital city poses new challenges for them to determine whether they can attain the information levels and master new knowledge and ideas in this fast-growing digital city.

Construction of the digital city and the training of management talents should be accelerated. The training of versatile and talented professionals in the digital city should receive greater attention to keep them studying and to obtain the latest technology and satisfy the growing demand of digital city construction. Digital Qianjiang is constantly updated in the continuous application to maintain the vitality of the platform.

### 5.4. Single-Investment Channel

Lack of application-oriented and compensation mechanism. In addition to the necessary technical support, a large amount of funding is needed to develop the digital city. Nevertheless, very little money can be devoted to the construction of a digital city, and most of the investments come from local revenue, which causes a single investment channel. If digital city geo-spatial frameworks have trouble ensuring sufficient money and compensation mechanisms and cannot even provide timely and accurate information services because of a failure to update, the further application in practical process will be hindered.

Expand finance channels and money input. The government should increase investment in the construction of digital city geo-spatial framework by bringing in a market competition mechanism, by organizing and encouraging active participation by enterprises and governments at all levels, and by coordinating relationships between both interests to work differently. It is the duty of government at all levels to invest in infrastructure construction and other basic public welfare projects for the digital city and to energetically strive for special national support. Corresponding preferential treatment should be given for some building projects with a cultural background, and local government should advise companies, groups, and people on investment under market mechanism circumstances, such as Public-private partnership (PPP). PPP models will play an important role in smart city financing

and funding. However, the success of the model will depend on how risk and reward are reallocated between public and private entities [31,32].

## 6. Conclusions

Digital Qianjiang completed the basic geographic information data collection, the multiple-scale basic geographic information database, the geographic information public platform, the basic geographic information distribution services, and several industry applications of geographic information systems, and it has improved the geographic information management and distribution services organization system. Digital Qianjiang established a real-time, public, unified geographic information space integration platform for the government and its departments to provide information on the city's unified, authoritative platform support. It promotes the full use of information sharing and data resources, explores the model of application of the framework of geo-spatial infrastructure, improves the scientific and working efficiency of government decision-making, reduces repeated investment, improves the scientific management level of the enterprise for the general public, provides the convenience of production and life, and allows for continuous emergency response and the construction of new socialist countryside to provide timely and reliable mapping protection. It has to be acknowledged that the smooth construction of digital Qianjiang is closely related to the fact that it is the first digital city. The leaders attached great importance to the relationship, which not only guaranteed the sufficient funds, but also provided advanced technologies. However, the department barriers and new technology requirements of data sharing are still the biggest bottleneck of digital city development.

The demonstration and leading role reflected in (1) establishment of a national, provincial, and city three-level co-construction and sharing mechanism, (2) establishment of a digital city technology safeguard mechanism, (3) participation of departments in linkage mechanism, and (4) accumulation of experience for an underdeveloped economy for the practice of small and medium-sized cities to carry out the construction of a digital city. We also found that Digital Qianjiang is likely to face great challenges—such as data acquisition and updating, sharing mechanisms and standard unification, single investment channel, service platform and application service extension limited, and personnel issues. These problems will be gradually highlighted in the construction and application of the digital city geo-spatial framework and must be effectively addressed to promote the healthy development of a digital city.

Dameri believed that the output will be a set of instruments used by different cities and governments to define and measure digital city performance; the collection may include data sets and data collection, indicators collected in indexes, policy making recommendations, project deliverables, and performance measurement frameworks [33]. We realize that the transformation and upgrading of the digital city to the smart city is an inevitable trend of the social development. The cores of the digital Qianjiang are the basic database building and management, data analysis and presentation and data sharing and services. The smart Qianjiang will be more focused on comprehensive data collection, dynamic monitoring (real-time data), interoperability of multi-information system, effective business management system, big data analysis and knowledge generation and prediction and support decision support. Thus, the construction of smart Qianjiang space-time information cloud platform becomes the inexorable trend of the upgrading for the digital Qianjiang geo-spatial framework.

**Author Contributions:** Supervision, M.Z. (Ming Zhang) and M.Z. (Ming Zhong); Writing—review & editing, H.W.

**Funding:** This study received partial support from the State Key Program of National Natural Science of China (Grant No. 41730640) and the 12th Five Year National Tech Support Grant for Smart Urbanization (Grant No. 2015BAJ05B00).

**Conflicts of Interest:** The authors declare no conflicts of interest.

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
