# Peer review of "Opportunities and Challenges for the Construction of a Smart City Geo-Spatial Framework in a Small Urban Area in Central China"

_smartcities, doi:10.3390/smartcities2020016_

Round 1
Reviewer 1 Report
The "Opportunities and Challenges for the Construction of a Digital City Geospatial Framework in a Small Urban Area in Central China" manuscript proposes to discuss the Chinese government's initiative in the development of cities by introducing the use of technologies by making them digital or intelligent. The theme is interested in getting the reader to know about this project. The text was well written, but there are some considerations. The authors need to pay attention to maintaining some standard, sometimes using the term "geo-spacial" with hyphens (line 17) and in others without the hyphen as "geospatial" (in title and line 52). In the phrase "makes it xxx" of line 170, some information seems to have been forgotten. The objectives of the article are clear, but its methodology is not. As a suggestion, the authors could create a section or sentence to present the methodology used since it was subjective. Many concepts and data are presented without their respective sources and references. The authors also need to be wary of the data presented as in line 64 and said that "the City of Qianjiang had a total population of 962,000", but in line 162 and said "total population of one million." Perhaps 38,000 is not a significant number for the Chinese standards, but in other parts of the world, there are cities with less than 10,000 inhabitants as in Brazil and Canada. Session 2.1 needs to be expanded, perhaps including a timeline of the development of digital cities around the world. The conclusion can be better explored and developed.
Author Response
Response to reviewer’s comments
We would like to thank the reviewer for the thoughtful review of this paper. All changes have been marked in red in the revised manuscript.
Reviewer: 1
Comments to the Author
The "Opportunities and Challenges for the Construction of a Digital City Geospatial Framework in a Small Urban Area in Central China" manuscript proposes to discuss the Chinese government's initiative in the development of cities by introducing the use of technologies by making them digital or intelligent. The theme is interested in getting the reader to know about this project. The text was well written, but there are some considerations.
The authors need to pay attention to maintaining some standard, sometimes using the term "geo-spacial" with hyphens (line 17) and in others without the hyphen as "geospatial" (in title and line 52).
Response:
All the "geospatial" terms have been changed to "geo-spacial".
In the phrase "makes it xxx" of line 170, some information seems to have been forgotten.
Response:
This sentence has been revised.
The regional transportation system in Qianjiang includes China National Highway 318 and the Shanghai-Chengdu expressway run across the city from east to west; two secondary roads Qianjiang and Xiangyue run north to south, and Qianjiang is the pilot city for national road network construction.
The objectives of the article are clear, but its methodology is not. As a suggestion, the authors could create a section or sentence to present the methodology used since it was subjective.
Response:
Thanks for the suggestion. See line 190 shows the relative explanation.
Geographic Information Data System is the core of the geo-spatial framework. It contains surveying and mapping, basic geographic information data, service-oriented product data, management system, and the technical supporting environment.
Many concepts and data are presented without their respective sources and references. The authors also need to be wary of the data presented as in line 64 and said that "the City of Qianjiang had a total population of 962,000", but in line 162 and said "total population of one million." Perhaps 38,000 is not a significant number for the Chinese standards, but in other parts of the world, there are cities with less than 10,000 inhabitants as in Brazil and Canada.
Response:
Thanks for the suggestion. The total population of Qianjiang was corrected to 962,000.
Session 2.1 needs to be expanded, perhaps including a timeline of the development of digital cities around the world. The conclusion can be better explored and developed.
Response:
Thanks for the suggestion. This session has been revised in the revised MS.

Reviewer 2 Report
General comments:
The text should deepen the bibliographic research about smart cities, their differences between digital and intelligent cities, as well as the concept of sustainable city. The text does not clarify which concept is actually related to the introduction. The concept of geo-referencing and geo-spatiality should be deepened as well as existing tools. It is an important contribution to the history of the case study chosen.
- Lines 32-33: cite source from where the number of cities was withdrawn;
- Lines 48-50: what scientific basis corroborates this view of smart cities?
- Line 64: there are studies that support a global urbanization movement in cities, to quote them.
- Lines 101-102: to include the role of the internet of things sensors for the development of the digital layer;
- Lines 108-109: there are more types of application than the three mentioned, please deepen the research;
- Line 131: cite in the references the statistical source searched;
- Line 152: the article has more than one author;
- Line 251: Are there scientific documents and authors that confirm this perception? If so, quote them.
- Line 259: Is this data current?
- Line 282: cite in the references the statistical sources;
- Lines 333 and 334: the goal was or is it still?
- Lines 358-359: Are the problems still persisting?
- Lines 369-373: Based on which authors can the statements be made?
- Line 390: explain what subjects are part of a digital city;
- Lines 419-421: the topic of PPP can be better explored, there is much literature on this source of funding. As well as international loans for financing the Smart Cities;
Final considerations: Conclusion data should be supported by extensive supporting documentation, data presented in the text do not allow for such generalization, nor can single-case studies be generalized.
Studies on the methodology of the case study need to be improved from the scientific point of view. From the point of view of the research approach at no time was the type of qualitative study approached.
How was the research protocol built? What were the primary and secondary goals? Has there been a systematic study of the primary documents? Which papers make theoretical support? How was the process of defining the unit of analysis? Were there people interviewed? What sources of evidence have been used? What was the period of the research in question? In order to have internal validity it is necessary to have documented the analysis of the multiple sources of evidence.
Author Response
Response to reviewer’s comments
We would like to thank the reviewer for the thoughtful review of this paper. All changes have been marked in red in the revised manuscript.
Reviewer: 2
Comments to the Author
The text should deepen the bibliographic research about smart cities, their differences between digital and intelligent cities, as well as the concept of sustainable city. The text does not clarify which concept is actually related to the introduction. The concept of geo-referencing and geo-spatiality should be deepened as well as existing tools. It is an important contribution to the history of the case study chosen.
- Lines 32-33: cite source from where the number of cities was withdrawn;
Response:
The relative reference has been added in revised MS.
- Lines 48-50: what scientific basis corroborates this view of smart cities?
Response:
Thanks for the suggestion. This part tried to explain the main differences and development relationship between the “Digital cities” and “Smart cities”.
- Line 64: there are studies that support a global urbanization movement in cities, to quote them.
Response:
Thanks for the suggestion. The global urbanization had not been mentioned in this part.
- Lines 101-102: to include the role of the internet of things sensors for the development of the digital layer;
Response:
Thanks for the suggestion. See revised MS lines 101-102, this sentence has been revised.
The rapid development of big data, cloud computing, internet of things and artificial intelligence technology has led a transition from digital city to smart city.
- Lines 108-109: there are more types of application than the three mentioned, please deepen the research;
Response:
Thanks for the suggestion. In fact, there are many applications, however in this part three main types had been selected and discussed. The whole part has been revised carefully and presented in the revised MS.
- Line 131: cite in the references the statistical source searched;
Response:
Thanks for the suggestion. The relative reference has been added in the revised MS.
- Line 152: the article has more than one author;
Response:
“Author” has been changed to “authors”.
- Line 251: Are there scientific documents and authors that confirm this perception? If so, quote them.
Response:
The benefits of Digital Qianjiang had been mentioned in the Government working report, and the relative reference has been added in the revised MS.
- Line 259: Is this data current?
Response:
Yes, the detailed data was presented in the reference 27, and we also added the relative reference here in the revised MS.
- Line 282: cite in the references the statistical sources;
Response:
The relative reference has been added in the revised MS.
- Lines 333 and 334: the goal was or is it still?
Response:
For Qianjiang city, this purpose has been achieved, however there are still lots of work should to be considered.
- Lines 358-359: Are the problems still persisting?
Response:
Yes, but the construction of a digital urban geo-spatial framework will solve this problem gradually.
- Lines 369-373: Based on which authors can the statements be made?
Response:
This part has been revised in the revised MS.
A sound information management system, information development planning, and service standards should be established in areas like the digital city’s construction framework, system construction, and service pattern. They should also standardize and integrate construction of geo-spatial frameworks and intensify the top-level design of the digital city.
- Line 390: explain what subjects are part of a digital city;
Response:
Computers, GIS, the internet, software, management, society, humanity, and other subjects are the part of a digital city.
- Lines 419-421: the topic of PPP can be better explored, there is much literature on this source of funding. As well as international loans for financing the Smart Cities;
Response:
The relative references have been added in the revised MS.
Final considerations: Conclusion data should be supported by extensive supporting documentation, data presented in the text do not allow for such generalization, nor can single-case studies be generalized. Studies on the methodology of the case study need to be improved from the scientific point of view. From the point of view of the research approach at no time was the type of qualitative study approached. How was the research protocol built? What were the primary and secondary goals? Has there been a systematic study of the primary documents? Which papers make theoretical support? How was the process of defining the unit of analysis? Were there people interviewed? What sources of evidence have been used? What was the period of the research in question? In order to have internal validity it is necessary to have documented the analysis of the multiple sources of evidence.
Response:
“Digital Qianjiang” is cooperatively built by National administration of surveying, mapping and geoinformation, Hubei administration of surveying, mapping and geoinformation and Qianjiang city government. Invested funds from National administration of surveying , mapping and geoinformation is mainly used for top-level design, high resolution aerial photography and platform software; Funds invested by Hubei administration of surveying , mapping and geoinformation is mainly used for geographic information data services with medium and small scale, database construction, public service platform construction, etc; funds invested by Qianjiang city government is mainly used in the geographic information data acquisition with city large proportion, industry application service system, hardware platform, and environment, etc. The mayor of Qianjiang city was interviewed by Xinhuanet, and the relative reference had been listed in Ref 28. Digital Qianjiang were approved by National administration of surveying, mapping and geoinformation on October 20, 2005, to the time of the project has been completed after the acceptance concerned by National administration of surveying, mapping and geoinformation in July 2009.
In summary, thanks for all the suggestions. These suggestions will be considered carefully, and the relative work will be carried out next.
Round 2
Reviewer 2 Report
The text should deepen the bibliographical research on intelligent cities, their differences between digital and intelligent cities, as well as the concept of sustainable city. The text does not clarify which concept is actually related to the introduction. The concept of geo-referencing and geospatiality should be deepened, as should existing tools.
It is an important contribution to the history of the chosen case study and has not yet been carried out in the article.
Answer: The introduction did not bring the requested advances or the deepening of the subjects related above. Increase the amount of theoretical foundation (citations and concepts). See lines 45-51. A digital city is not a smart city.
Lines 32-34: There are over 1,000 smart cities in the world that have been launched or under construction [1]. Europe, North America, Japan and South Korea are the main regions of 34 smart cities.
Answer: Insert the data sources that corroborate with the highlight of each country, if you have them.
Lines 36-37: In addition, large companies like Cisco and IBM continue to deepen this market, as Cisco 2017 announces $ 1 billion program for smart cities.
Answer: Insert source that informs CISCO and IBM program. All that is to be used as reference must have the author of the original text quoted.
Final considerations: characterize the case study from a methodological point of view. See in Yin, Robert K. Case study research: design and methods.
From reading the case study requirements, the text should be revised to include the conclusion supported by extensive documentation, the data presented in the text do not allow such generalization, nor can single-case studies be generalized.
Studies on the methodology of the case study need to be improved from the scientific point of view. From the point of view of the research approach, at no time the type of qualitative study was approached.
The following questions were not answered in the text of the article:
How was the research protocol built?
What were the primary and secondary objectives?
Has there been a systematic study of the primary documents?
Which documents are supported by theory?
How was the process of defining the unit of analysis?
Were there people interviewed?
What sources of evidence were used?
What was the period of the research in question?
In order to have internal validity, it is necessary to document the analysis of the multiple sources of evidence.
Author Response
MS SmartCities-514480 Revision 2
Response letter
Many thanks to the reviewer for the valuable and constructive comments and suggestions! Below we provide detailed information (text in boldface) on our responses to the reviewer comments and suggestions.
The text should deepen the bibliographical research on intelligent cities, their differences between digital and intelligent cities, as well as the concept of sustainable city. The text does not clarify which concept is actually related to the introduction. The concept of geo-referencing and geospatiality should be deepened, as should existing tools.
It is an important contribution to the history of the chosen case study and has not yet been carried out in the article.
Answer: The introduction did not bring the requested advances or the deepening of the subjects related above. Increase the amount of theoretical foundation (citations and concepts). See lines 45-51. A digital city is not a smart city.
Response:
Thanks to the reviewer’s suggestion!
We accordingly rewrote the Introduction part of the manuscript. We stated explicitly the difference and relationship between Digital City and Smart City and interpreted that Digital City was the first phase of Smart City development. To support the statement, we provided key references and statistics on digital/smart cities (please refer to the Introduction part for detail).
With respect to the reviewer’s suggestion, we decided not to enter discussion on the concept of sustainable city. We felt that the topic was beyond the scope of this manuscript and including it might distract the main interest of the manuscript. Similarly, we did not talk about the concept of geo-referencing and geospatiality in the Introduction part because, we believed, they were very technical and could not be explained with clarity without taking a substantial amount of Introduction space; we were concerned that detailed technical discussion in Introduction could confuse the reader on the main purpose of this manuscript.
Answer: Insert the data sources that corroborate with the highlight of each country, if you have them.
Response:
Done!
The related reference has been added in the revised MS.
Deloitte, Available from: https://www2.deloitte.com/cn/zh/pages/public-sector/articles/super-smart-city.html
Lines 36-37: In addition, large companies like Cisco and IBM continue to deepen this market, as Cisco 2017 announces $ 1 billion program for smart cities.
Answer: Insert source that informs CISCO and IBM program. All that is to be used as reference must have the author of the original text quoted.
Response:
Done!
The relative reference has been added in the revised MS.
Final considerations: characterize the case study from a methodological point of view. See in Yin, Robert K. Case study research: design and methods.
From reading the case study requirements, the text should be revised to include the conclusion supported by extensive documentation, the data presented in the text do not allow such generalization, nor can single-case studies be generalized.
Studies on the methodology of the case study need to be improved from the scientific point of view. From the point of view of the research approach, at no time the type of qualitative study was approached.
The following questions were not answered in the text of the article:
How was the research protocol built?
What were the primary and secondary objectives?
Has there been a systematic study of the primary documents?
Which documents are supported by theory?
How was the process of defining the unit of analysis?
Were there people interviewed?
What sources of evidence were used?
What was the period of the research in question?
In order to have internal validity, it is necessary to document the analysis of the multiple sources of evidence.
Response:
Thanks to the reviewer for raising these important questions!
In response to the questions, we added a new section to the manuscript, “3. Study Method and the Case of Digital Qianjiang.”
We hope we have addressed the methodological concerns that the reviewer had. Many thanks to the reviewers’ great suggestions!
